# Sex Differences in the Effect of Alcohol Drinking on Myelinated Axons in the Anterior Cingulate Cortex of Adolescent Rats

**DOI:** 10.3390/brainsci9070167

**Published:** 2019-07-16

**Authors:** Elizabeth R. Tavares, Andrea Silva-Gotay, Wanette Vargas Riad, Lynn Bengston, Heather N. Richardson

**Affiliations:** 1Department of Psychological and Brain Sciences, University of Massachusetts, Amherst, MA 01003, USA; 2Neuroscience and Behavior Graduate Program, University of Massachusetts, Amherst, MA 01003, USA; 3BGB Group 462, Broadway, New York, NY 10013, USA

**Keywords:** adolescence, alcohol use disorder, rats, operant self-administration, sex differences, females, males, prefrontal cortex, anterior cingulate cortex, myelin, nodes of Ranvier, contactin-associated protein 1

## Abstract

Cognitive deficits associated with teenage drinking may be due to disrupted myelination of prefrontal circuits. To better understand how alcohol affects myelination, male and female Wistar rats (*n* = 7–9/sex/treatment) underwent two weeks of intermittent operant self-administration of sweetened alcohol or sweetened water early in adolescence (postnatal days 28–42) and we tested for macro- and microstructural changes to myelin. We previously reported data from the males of this study showing that alcohol drinking reduced myelinated fiber density in layers II–V of the anterior cingulate division of the medial prefrontal cortex (Cg1); herein, we show that myelinated fiber density was not significantly altered by alcohol in females. Alcohol drinking patterns were similar in both sexes, but males were in a pre-pubertal state for a larger proportion of the alcohol exposure period, which may have contributed to the differential effects on myelinated fiber density. To gain more insight into how alcohol impacts myelinated axons, brain sections from a subset of these animals (*n* = 6/sex/treatment) were used for microstructural analyses of the nodes of Ranvier. Confocal analysis of nodal domains, flanked by immunofluorescent-labeled contactin-associated protein (Caspr) clusters, indicated that alcohol drinking reduced nodal length-to-width ratios in layers II/III of the Cg1 in both sexes. Despite sex differences in the underlying cause (larger diameter axons after alcohol in males vs. shorter nodal lengths after alcohol in females), reduced nodal ratios could have important implications for the speed and integrity of neural transmission along these axons in both males and females. Alcohol-induced changes to myelinated axonal populations in the Cg1 may contribute to long-lasting changes in prefrontal function associated with early onset drinking.

## 1. Introduction

Alcohol binge drinking is highly prevalent in teenagers [1] and is associated with high costs to global health and economy [2]. Alcohol consumption contributes to a number of social issues, including increased violent behaviors, damage to personal relationships, law enforcement violations, interference with education, and risk for additional substance abuse [3]. During adolescence, brain regions such as the prefrontal cortex are still undergoing development, characterized by decreases in gray matter and increases in white matter [4,5]. Human studies have shown correlations between adolescent alcohol consumption and reductions in frontal white matter. Binge drinking adolescents had a significant reduction in functional anisotropy (white matter integrity) in major white matter frontal pathways when compared to controls [6,7,8]. These effects, in turn, are associated with a higher risk of developing alcohol dependence in adulthood [9,10]. While these human studies allow us to observe correlations between alcohol and myelin, they cannot directly test causal relationships.

Animal models of alcohol exposure have been used to test the causality of the links between early onset drinking and reduced myelin integrity in humans. Intermittent exposure to alcohol via intragastric delivery has been shown to alter myelin-related gene expression in adolescent male and female mice [11]. High doses of alcohol administered intermittently via intragastric delivery during adolescent development in male rats causes object recognition memory deficits in adulthood, as well as reduced axial diffusivity—an index of myelin integrity—in the neocortex, hippocampus, and cerebellum [12]. Likewise, intermittent exposure to high doses of alcohol via intraperitoneal injections early in adolescence reduces glial cell number in the medial prefrontal cortex (mPFC) in adulthood in male rats [13]. Interestingly, this same dose, route, and pattern of alcohol exposure during adolescence does not elicit measurable changes in glial cell number in the mPFC of female rats, but it has been shown to cause long-term effects on myelin in the prefrontal cortex, including downregulated myelin proteins and aberrant compaction of myelin sheaths in adult female mice [14]. These studies demonstrate the ability of high doses of alcohol to impact myelin proteins and white matter, but it should be noted that the exposure to alcohol was involuntary. This is an important factor to consider, as we have found that binge-like alcohol injections during adolescence can lead to lower baseline levels of alcohol drinking in adulthood, whereas drinking in adolescence can lead to higher levels of relapse after short periods of abstinence in adulthood [15]. It is therefore beneficial to study the effects of voluntary drinking on myelinated axons to gain insight into why early onset alcohol use is linked to white matter deficits and heightened alcohol use disorder risk in adulthood in humans [6,7,8,9,10].

Most of what we know about the effect of voluntary drinking on myelin comes from studies in adult animals. Five months of alcohol consumption with 10% v/v ethanol as the only source of liquid causes a reduction of myelin basic protein immunoreactivity in the prefrontal cortex, a downregulation of proteins and mRNAs involved in myelination in the cortex, an increase in oligodendrocyte cell death in the mPFC, and disruptions in myelin compaction in the corpus callosum and cortex of adult mice [16]. Another study in adult rats showed that one year of alcohol drinking reduces corpus callosum thickness, area, and myelin thickness in adult rats exposed, as compared to controls [17]. These studies highlight the effect of extensive alcohol consumption in adulthood on myelin; however, we have found that a much shorter period of exposure impacts myelin in adolescent animals, suggesting heightened vulnerability of this developmental period. Just two weeks of operant self-administration of alcohol during early adolescence reduces myelinated fiber density in the dorsal subdivision of the mPFC (anterior cingulate, Cg1) of male rats [18]. Moreover, these effects of adolescent alcohol persist into adulthood, evidenced by reduced white matter in the prefrontal cortex—an effect that related to a higher propensity to relapse to alcohol after short periods of abstinence. These results demonstrate that voluntary binge drinking during adolescence can have severe consequences on prefrontal myelination, and these changes can be long-lasting. However, the previously mentioned rodent studies highlight sex differences in alcohol-induced alterations in glia cells [13], suggesting that alcohol could affect myelin differently in males and females. The aim of the present study was to test for sex differences in the effect of voluntary alcohol drinking during early adolescence on myelinated axons in the Cg1. We found sex differences in the effect of adolescent alcohol drinking on myelin at the macro- and microstructural levels, with males showing heightened sensitivity to alcohol. Our data altogether suggest that myelinated axons in the prefrontal cortex are sensitive to alcohol during adolescence and could potentially have functional implications for both males and females. These findings provide insight into a neuropathological mechanism by which early onset drinking could lead to long-term mental health risks in adulthood.

## 2. Materials and Methods

### 2.1. Animals

The brains from 47 Wistar rats (24 males and 23 females) were used in the current study. Additional information about these animals (details about training, operant equipment, body weight, etc.) can be found in [19]. Pups from a mix of litters were shipped with nursing dams on postnatal day (PD) 18 from Charles River (Wilmington, MA, USA) and were housed together until PD 21. After weaning, PD 21 rats were housed in same-sex groups of three. During the training period, rats were handled for a minimum of 5 minutes per day in order to acclimate them to human contact, thereby reducing potentially confounding handling stress during the experimental period. Food and water were available ad libitum and animals were kept on a 12-hour light/12-hour dark cycle. All procedures were approved by the Institutional Animal Care and Use Committee and performed in accordance with the National Institutes of Health Guide for the Care and Use of Laboratory Animals.

### 2.2. Experimental Design

The experimental design of this study is shown in Figure 1. Beginning on PD24, male and female rats were examined for external signs of pubertal maturation, indexed by vaginal opening in females [20,21] and separation of the balano-preputial gland (“preputial separation”) in males (Figure 1a) [22]. Rats self-administered sweetened alcohol (“alcohol”), sweetened water (“control”), or remained naive to the operant training/sweetened water (“naive”) for two weeks during early adolescence (PD 28–42, *n* =7–9/sex/treatment, Figure 1b,c). Animals were randomly assigned to the operant and naive groups. Training behavior was then used to ensure control and alcohol groups were balanced for lever-pressing behavior before self-administration began on PD 28. On PD 43 (~12 hours after the 1st bout and 4 hours after the 6th bout of the last overnight self-administration session) animals were perfused and the brains were further processed for future use in Experiments 1 and 2 (Figure 1d). For Experiment 1, all of the brain tissue (*N* = 47, *n* = 7–9/sex/treatment groups: naive, control, alcohol) was used for myelin labeling with Black Gold II for microscopic analysis of myelin density (Figure 1e). For Experiment 2, a subset of brain tissue (*N* = 24, *n* = 6/sex/treatment groups: control, alcohol) was used for immunofluorescent labeling of the nodes of Ranvier to analyze nodal densities and dimensions (Figure 1f). As the naive group was not different from the control group in overall myelin density readout, this group was not included in Experiment 2 as a means of refining the labor-intensive nodal analyses.

### 2.3. Operant Training and Adolescent Alcohol Exposure

The timeline of operant self-administration and measures of alcohol intake are shown in Figure 1a–c. The original report shows the cumulative g/kg alcohol consumption in the alcohol groups and of cumulative g/kg glucose in the alcohol and control groups [19]. The current study provides additional analyses of drinking behavior (Figure 1b,c). Beginning on PD 25, animals were trained to lever press for sweetened water (3% glucose/0.125% saccharin/tap water) on a fixed-ratio 1 (FR1) schedule in operant boxes as previously described [15,19]. Three days later, operant-trained rats were assigned to either the control condition (remained on sweetened water) or the alcohol condition (switched to sweetened alcohol; 10% v/v ethanol/sweetened water). The treatment groups were balanced based on training operant responses for sweetened water. Operant sessions during experimental binge periods consisted of six 30-minute bouts divided by 60-minute timeout periods when levers were retracted, and the sweetened solutions were unavailable. To ensure voluntary consumption of alcohol, animals were fed ad libitum, meaning they had free access to rat chow and water throughout all phases of the experiment. Animals were tested individually in separate operant boxes during experimental binge periods and all lever press data were recorded using Med-PC IV software (Med Associates Inc., Latham, NY, USA).

### 2.4. Perfusions and Brain Sectioning

All animals were deeply anesthetized with chloral hydrate and then intracardially perfused with 4% paraformaldehyde the day after the two-week binge period ended (PD 43). Brains were post-fixed for 4 hours, and then put into 20% sucrose for 24–48 hours. Brains were then snap frozen by submerging briefly in isopentane (2-methylbutane; Thermo Fisher Scientific, catalogue #03551-4, Fair Lawn, NJ, USA) cooled with dry ice and then stored at −80 °C until sectioning. Serial coronal sections (35 µm thickness) were collected using a frozen microtome and stored in cryoprotectant (50% 0.1 M Phosphate Buffered Saline (PBS), 30% ethylene glycol, and 20% glycerol) at 20 °C until further processing for Experiments 1 and 2.

### 2.5. Experiment 1: Myelin Labeling and Microscopic Analyses of Myelin Density in the Cg1

The gold phosphate complex Black Gold II was used to label myelin sheaths in every 10th brain section [24]. Labeled sections were then mounted on glass slides, coverslipped, and digitally scanned at 20× using Aperio Digital Pathology Slide Scanner (Leica Biosystems, Buffalo Grove, IL, USA). Sections that were 2.2 mm anterior to Bregma [25] were used for analysis of myelin density in cortical layers II–V in the Cg1. Aperio ImageScope software (Leica Biosystems Imaging Inc., Vista, CA, USA) was used to quantify myelinated fiber density by thresholding the images. In Figure 1d, a grayscale image of a section with Black Gold II-labeled fibers highlighted in red was used to help illustrate how myelinated fibers can be visualized and the overlaying pixels can be quantified (Figure 1e). The percentage of area covered by myelinated fibers over the total area was calculated to quantify myelinated fiber density. All analyses for Experiment 1 were conducted by investigators blind to the sex and treatment conditions of the experiment.

### 2.6. Experiment 2: Labeling and Confocal Analysis of the Node of Ranvier in the Cg1

#### 2.6.1. Immunofluorescent Labeling

Every 10th brain section was sorted and the antibodies described below were used to immunofluorescently label the nodal domains on myelinated axons. Free-floating tissue sections were washed in 0.1 M Phosphate Buffered Saline (PBS) followed by rinses with PBS with 0.2% Triton X (PBS-Tx). Endogenous fluorescence was quenched with 50 nM ammonium chloride in PBS-Tx. Following additional washes, sections were blocked for non-specific binding using 5% normal horse serum in PBS-Tx for one hour and then incubated overnight in primary antibodies. The following primary antibodies were used for visualization of the nodal, paranodal, and juxtaparanodal regions, respectively: rabbit anti-contactin-associated protein (Caspr) (Abcam, cat. # ab34151, 1:1000) and mouse anti-Kv1.2 (Neuromab, cat. # 75-008, 1:1200). Following overnight incubation (approximately 19 hours), tissue was rinsed in PBS-Tx and incubated in secondary antibodies (2% Bovine Serum Albumin (BSA), 5% Normal Horse Serum (NHS) in PBS-Tx; goat anti-rabbit Alexa Cy3 (Jackson Immuno, 1:300), biotinylated horse anti-mouse (Vector Laboratories, 1:200)) for 2 hours at room temperature. Following secondary antibody incubation, tissue was rinsed again in PBS and then incubated one final time for further amplification of the Kv1.2 signal (PBS-Tx; Cy2-conjugated Streptavidin (Jackson Immuno, 1:2000)) for 1 hour at room temperature. After the final incubation was complete, tissue was washed in PBS and all sections were mounted onto glass slides and dried before coverslipping with DPX mounting media (Cat. # 06522, Sigma-Aldrich, St. Louis, MO, USA). All fluorescently labeled samples were prepared, mounted, coverslipped, and stored under red-light and low-light conditions to minimize bleaching effects.

#### 2.6.2. Confocal Imaging Acquisition and Nodal Analyses

All images were obtained using a Nikon A1 system confocal microscope and NIS Elements Advanced Research software (Nikon Instruments Inc, Melville, NY, USA). For each sample, images were taken in layers II/III of the Cg1 (Figure 1f). We limited the sampling area to these layers to overcome the technical and labor-intensive challenges of this analysis. We have initially focused on layers II/III because the basolateral amygdala is still sending afferents to these layers during adolescence and into early adulthood [26], potentially making these developing fibers more vulnerable to the effects of alcohol. Images were acquired using an Apo 60× oil immersion objective (Numerical Aperture = 1.4) at Nyquist sampling. For each animal, 2–3 z-stacks were taken in each hemisphere, for a total of 4–6 stacks available per animal for analyses. After collecting z-stack images at 60×, 10× images were collected and stitched together to reconstruct each hemisphere. This was used to confirm sampled regions were consistent in all subjects. When necessary, adjustments were made and new images were collected to ensure sampling locations were matched between subjects.

For nodal count measurements, four z-stack images (2 per hemisphere) of equal volume (60.57 × 60.57 × 3.15 um^3^) were analyzed per animal using NIS Elements Advanced Research software (Nikon Instruments Inc, Melville, NY, USA). Counts were obtained manually—each node was defined as an associated ‘Caspr pair’ with relative symmetry, and nodes were marked with region of interest (ROI) boxes in the NIS Elements three-dimensional analysis software.

For nodal ratio measurements, a total of 40 nodes were analyzed per animal, obtained from 4–6 z-stacks per animal (the average number of z-stacks did not differ across groups). Two measurements were obtained at each node of interest (nodal length and nodal diameter) using a line drawing tool to measure Feret’s diameter. Regions of interest (ROIs) were saved to the document the nodes that had been measured. Nodal length and nodal diameter measurements were then used to calculate nodal ratios (nodal length/nodal diameter). All analyses for Experiment 2 were conducted by investigators blind to the sex and treatment conditions of the experiment.

### 2.7. Statistical Analyses

The age of pubertal onset was analyzed using between-subjects two-way ANOVA, with sex and treatment as between-subject variables. Daily average alcohol intake (g/kg) was analyzed using a mixed-model ANOVA with sex as a between-subject variable and operant day as a within-subject variable to test for escalation. Total alcohol intake (g/kg) over the two-week treatment period was analyzed by one-way ANOVA to compare male and female intake. Alcohol self-administration by bout (g/kg) was analyzed using a within-subjects two-way ANOVA, with bout and operant day as within-subject variables. Myelin density and nodal counts were analyzed using between-subjects two-way ANOVA, with sex and treatment as between-subject followed by Tukey multiple comparisons. Two-sample Kolmogorov–Smirnov test was used to compare nodal ratio, length, and diameter distributions between control and alcohol groups. Statistical significance was defined as *p* ≤ 0.05 using two-tailed tests. Wherever appropriate, data are presented as mean ± SEM. Statistical analyses were performed using the R statistical software package (open source from https://www.r-project.org) [27].

## 3. Results

### 3.1. Sex Differences in Pubertal Maturation

There was a main effect of sex on pubertal onset (F_(1,30)_ = 259.14, *p* < 0.001), with females showing external signs of sexual maturation significantly earlier than males—a finding that is consistent with the literature [28]. The data are displayed in Figure 1a as % pubertal curves to show that females showed signs of pubertal maturation at the beginning of the drinking period, while males remained pre-pubertal for another 7–10 days. No effect of treatment was detected (*p* = 0.4).

### 3.2. Drinking Patterns Were Similar in Males and Females

Figure 1 shows self-administration of alcohol (g/kg) within each 30-minute bout of the 6 bouts per overnight session (b), average intake per day (c), and total intake over the 2-week exposure period (c, inset bar graph). As previously reported, all groups consumed a similar amount of glucose, and there were no sex differences in the total amount of alcohol consumed in the binge groups ([19], Figure 1c). The lack of sex differences at this early adolescent time period is a consistent finding in our lab, which is likely because sex differences emerge later in adolescence [29] and continue into adulthood [30,31]. Here we show that male and female rats self-administered more alcohol in their first bout compared to the last bout of each day (males: F_(1,5)_ = 36.62, *p* = 0.002; females: F_(1,5)_ = 117.4, *p* = 0.0001, within-subjects two-way ANOVA, Figure 1b). This evidence of ‘front-loading’ when the alcohol is first introduced is consistent with other models of alcohol drinking in rodents [32,33,34,35].

### 3.3. Sex Differences in the Effect of Alcohol on Myelinated Fiber Density in the Cg1

We have shown previously that alcohol reduces the density of myelinated axons in the Cg1 of male rats [18]. As stated earlier, the males and females were run together for all stages of the study, e.g., self-administration, tissue processing, and imaging, thus allowing us to directly test for sex differences. A two-way ANOVA showed a significant interaction between sex and treatment (F_(2,21)_ = 4.72, *p* = 0.01, Figure 2). Post-hoc analyses showed that alcohol reduced myelin density in males, but not in females.

### 3.4. Alcohol Did Not Cause Measurable Changes in the Number of Nodes of Ranvier in Layers II/III of the Cg1 of Male and Female Rats

To test whether the possible effects of alcohol on females can be detected at the microstructural level of myelin, we first measured the density of nodes of Ranvier in layers II/III of the Cg1 in both males and females (Figure 3a,b). The measurement of nodal densities was captured by counting the number of ‘Caspr pairs’ in a total of four z-stacks per animal, with each stack occupying an equal volume within layer II/III of the Cg1 (60.57 × 60.57 × 3.15 μm^3^). The number of nodes was not significantly altered after adolescent alcohol drinking, and there was no significant interaction between sex and treatment (*p* > 0.05, Figure 3b). These data further suggest that alcohol either does not impact myelin in females, or alcohol’s effects are not detectable at the macrostructural level. To delineate between these two possibilities, we further explored the effect of alcohol on myelin microstructure by examining the nodal domain in male and female rats.

### 3.5. Nodal Ratios Were Reduced by Alcohol in Layers II/III of the Cg1 of Male and Female Rats

We first examined nodal length in relation to nodal diameter (Figure 3c), also known as nodal ratios [36,37]. A two-sample Kolmogorov–Smirnov test showed that nodal ratio distribution was shifted to the left in the Cg1 of both males (*p* = 0.001) and females (*p* = 0.001), suggesting that alcohol induces changes in nodal microstructure in both sexes (Figure 3d). There were no sex differences in nodal ratios in control animals (*p* = 0.15). We next investigated whether this shift in the distribution of nodal ratios was due to a decrease in nodal length or an increase in axon diameter.

### 3.6. Alcohol-Induced Decreases in Nodal Ratios Were Due to Larger Nodal Diameters in Males and Shorter Nodal Lengths in Females

Figure 4 shows the distribution of the nodal domain measurements obtained from 240 myelinated axons per treatment group in male and female rats. Nodal lengths were obtained by measuring the length of the gap between matched Caspr pairs at the nodes of Ranvier and nodal diameters were obtained by measuring the thickness of each matched Caspr pair (Figure 3c). Nodal lengths and diameters were not different in male and female control animals (*p* > 0.05). Alcohol shifted the distribution of axonal diameters to the right in males (*p* = 0.0004), indicating that there was a greater proportion of myelinated axons with large diameters at least at the nodes (Figure 4a, right graph). This alcohol-induced shift in distribution of axon diameters was not evident in females (*p* > 0.05). In females, alcohol shifted the distribution of nodal lengths to the left (*p* = 0.0002), indicating a greater proportion of myelinated axons with shorter nodes (Figure 4b, left graph). This effect was not observed in males (*p* > 0.05).

## 4. Discussion

Binge drinking of alcohol is highly prevalent in teenagers and is associated with increased risk of alcohol use disorder in adulthood [38]. Indices used to estimate poor myelin integrity such as lower fractional anisotropy have been observed in major white matter frontal pathways of binge-drinking teenagers [6,7,8]. The main objective of the present study was to use a rodent model of voluntary drinking to gain insight into how alcohol impacts myelin in the developing brain. After two weeks of intermittent exposure to operant alcohol self-administration, myelinated fiber density was reduced in the Cg1 of males [18], but not females (current study). This suggests that myelinated axons are more sensitive to alcohol in males, but females were not completely protected from alcohol’s effects. Confocal analyses of contactin-associated protein (Caspr) indicated microstructural effects of alcohol at the nodes of Ranvier that could disrupt the neurotransmission speed and integrity in prefrontal circuits in both males and females. Considering the macro- and microstructural changes we observed and what is known about demyelination/remyelination, we can conclude that drinking early in adolescence (1) has a more robust effect on myelinated axons in males, (2) likely causes demyelination followed by partial remyelination in males, and (3) alterations in the dimension of the nodal domain could result in functional changes in neurotransmission. These findings increase our understanding of mechanisms by which heavy drinking could impair cognitive abilities—possibly through reduced neurotransmission speed and efficacy among myelinated prefrontal pathways.

Several pieces of evidence suggest that alcohol drinking may have caused demyelination, followed by the initiation of remyelination of large diameter axons—at least in male rats (summarized in Figure 5). As de novo myelination of axons in the Cg1 extends into early adolescence [39], it is possible that the reduced myelin density observed in alcohol males reflects a delay of myelination of axons within this region. However, one prediction of a developmental delay in myelination would be a lower number of nodes. Instead, we show a trend of an increased number of nodes in the Cg1 of males rather than a decrease, despite reducing myelin density in this region. It is therefore conceivable that these prefrontal axons underwent demyelination, and remyelination was initiated before the end of the two-week binge drinking period. This hypothesis is based on evidence showing that during the course of exposure to demyelinating agents such as the copper chelator cuprizone, remyelination eventually kicks in and the two phases can overlap in time [40]. After the internodes (myelin segments) are removed during demyelination, they can be replaced by new oligodendrocyte processes but the replacement internodes are often thinner [41] and are shorter in length [42], resulting in an increase in the number of nodes, similar to the trend we saw in alcohol males. We also found that after alcohol, the population of myelinated axons shifted toward larger diameter axons. These results could indicate preferential demyelination of smaller axons, or alternatively preferential remyelination of larger axons following demyelination. While it is currently unknown whether smaller axons are more vulnerable to demyelination, it is known that larger axons are remyelinated first [40,43]. Regardless of the details of the demyelinating/remyelinating processes, the consequential changes in myelin microstructure could have major functional consequences, as shorter internodes can result in decreased conduction velocity [44].

While girls are often included in studies linking reduced prefrontal white matter to binge drinking in teenagers, boys usually comprise a higher proportion of the AUD sample [7,47,48]. Thus, it is difficult to have a large enough dataset to fully examine sex differences under identical alcohol exposure conditions, e.g., drink patterns, time since last drink, etc. For example, in one study that included 5 female and 9 male subjects with AUD, prefrontal white matter volume was found to be reduced in females, but surprisingly elevated in males compared to non-AUD controls [47]. While this finding in males may seem contrary to other reports in humans, e.g., [7,48], the average time since the last drink was 21 days (versus 9 days in female subjects). Based on our results showing that remyelination could possibly be initiated before drinking ends in males, it is conceivable that the white matter volume has rebounded during those three weeks of abstinence from alcohol in the teenage boys of that study.

Our results show that following the same level of alcohol exposure (drinking pattern and g/kg intake), prefrontal myelin is more negatively impacted in male rats compared to female rats. Alcohol drinking did not affect overall myelin density in females, unlike in males, and the number of nodes remained unchanged after alcohol exposure. This suggests that overall female myelin macrostructure was not affected by alcohol, or that alcohol caused demyelination but females were able to remyelinate efficiently enough to completely restore myelinated fiber density. However, results from the literature suggest that the latter hypothesis is unlikely; as discussed above, remyelination results in thinner and shorter internodes [41,42,45,49,50]. Thus, if myelin density is fully restored, it would have to be accompanied by more nodes. Remyelinating oligodendrocytes also tend to wrap axons that have already reached their mature size, which would cause a shift in the axon diameter distribution [40,43]. None of these scenarios is consistent with our data, suggesting that myelin density does not change in females because these axons do not undergo demyelination and remyelination to the same extent as males. Alternatively, it is also possible that remyelination dynamics are different in females at this stage. Further studies are required to completely rule out this hypothesis.

While alcohol may not affect overall myelin architecture in females, we did observe a shift in nodal length distribution to shorter nodes. As this change is not accompanied by a change in myelin density or number of nodes, we infer that existing nodes became more condensed. This phenomenon has been observed before in a chronic stress mouse model, where stress disrupted the distribution of proteins at the nodes and paranodes of Ranvier in the corpus callosum and resulted in shorter nodes [51]. The functional consequences of shorter nodes remain unclear. A mathematical model of myelinated axons suggests a decrease in nodal length is beneficial, with increasing conduction velocity as the nodal length decreases [52]. However, a different model suggests that there is an optimal nodal length, and anything too short would become suboptimal, as the number of sodium channels would decrease [37]. In addition, electrophysiological studies performed in transgenic Oligodendrocyte myelin glycoprotein knockout mice show that a shortening of nodal length results in a significantly lower conduction velocity than wild-type mice [53]. Therefore, the observed shift to smaller nodal gap lengths in females could imply impaired functionality. Indeed, exposure to high doses of alcohol via gavage can elicit working memory deficits in adult male and female rats [54]. We have found that high adolescent drinking levels predicted working memory deficits in adulthood in males [18]. It remains to be determined if this relationship is also observed in high drinking females.

Alcohol can interrupt the birth and differentiation of oligodendrocyte lineage cells [55]. It is possible that baseline sex differences in oligodendrocyte turnover could account for sex differences observed in the effects of alcohol on myelin. Here we show that adolescent females are in fact affected by alcohol, but they appear to be less sensitive than their male counterparts during this period. Different factors could account for the sex differences observed in the effects of alcohol on myelin macro- and micro-structure. Although our data seem to indicate that alcohol does not cause demyelination in females we are not able to rule out this hypothesis completely, as this is a single timepoint at the end of a chronic period of repeated alcohol exposure. If alcohol did, in fact, cause demyelination in females, we would have to assume that they were able to completely restore myelinated fiber density to normal levels. There is evidence suggesting that females would be able to remyelinate more efficiently than males. Males have a higher density of mature oligodendrocytes in white matter areas like the corpus callosum [56,57], yet females have more oligodendrocyte progenitor cells (OPCs) and a higher turnover of oligodendrocytes [56], which is important for remyelination [58,59,60]. As such, there would be a system in place to quickly respond to demyelination in females.

In addition to the sex differences discussed above, males and females also display measurable differences in maturation during the period of alcohol exposure in this study (PD28–42), including differences in the levels of circulating gonadal hormones and the age of pubertal onset. Circulating levels of gonadal hormones increase in females much earlier than they do in males, resulting in a much earlier age of pubertal onset, as shown in our data and supported by the literature [28]. Gonadal hormones, especially estradiol, have been shown to promote myelination and remyelination after a demyelinating injury in white matter regions [57]. In that same study, gonadectomized groups display more severe demyelination, suggesting gonadal hormones also have a protective effect against demyelination. In our sample, females already have high circulating levels of estradiol and progesterone, but testosterone levels in males do not increase to adult-like levels until after PD42 [28]. Thus, ovarian hormones could be protecting females from the full effects of alcohol by stimulating myelin basic protein expression in oligodendrocytes [61] to prevent demyelination after each alcohol insult.

## 5. Conclusions

Early onset drinking has been linked to increased mental health risk and alcohol use disorder [38], and the present findings in conjunction with previous studies suggest that alcohol may perturb myelination of axons in prefrontal circuits during adolescence to drive some of these increased risks. White matter continues to increase during adolescence [62] and alcohol use during this time has been related to poor white matter integrity [7,8,63,64]. Intragastric or intraperitoneal administration of high doses of alcohol has been shown to damage myelin sheaths and decrease expression of myelin genes and proteins in the prefrontal cortex of adolescent male and female mice [11,14,65]. Voluntary consumption of alcohol early in adolescence reduces myelinated fiber density in the Cg1 and the size of the anterior branches of the corpus callosum (corpus callosum-forceps minor, CC_FM_) in male rats [18]. In addition, the current findings indicate that alcohol changes nodal domain microstructure on the remaining myelinated axons in both male and female rats. These findings altogether highlight the heightened sensitivity of prefrontal axons to alcohol. One caveat of our study is that the myelin density changes observed in males includes layers II–V, but only layer II/III was analyzed at the microstructural level. While our findings indicate that alcohol alters nodal domains of myelinated axons within layer II/III of males and females, it is also possible that alcohol causes similar microstructural changes to axons in layer V. This is especially important because this drinking period overlaps with de novo myelination of the axons extending from the CC_FM_ through layer V of the Cg1, which is accompanied by a doubling of transmission speed [39]. This raises the possibility that alcohol reduces neural transmission speed and integrity by impacting prefrontal myelin, ultimately impairing functions dependent on a healthy prefrontal cortex.

## Figures and Tables

**Figure 1 brainsci-09-00167-f001:**
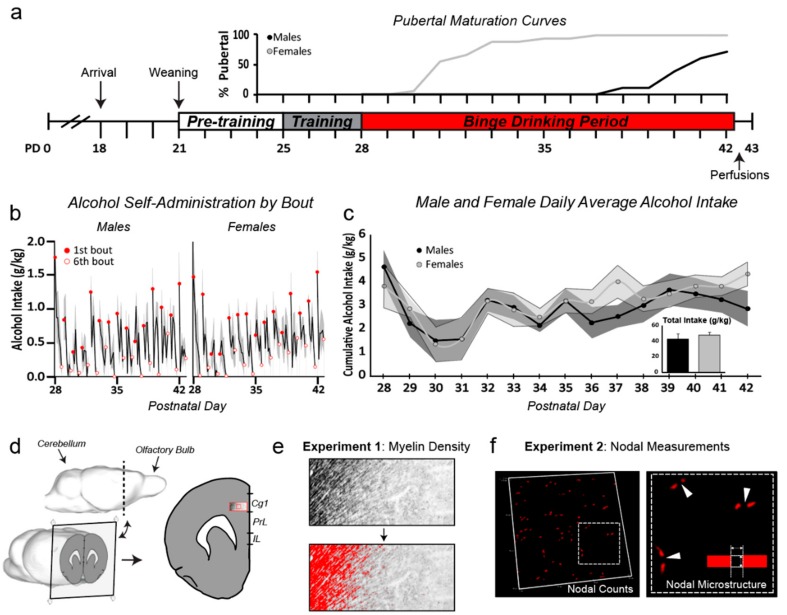
**Overview of experimental design, pubertal maturation, and alcohol self-administration.** (**a**) Timeline of operant self-administration and pubertal maturation curves (indexed by external signs of pubertal onset). Females were in a pre-pubertal state for a few days to one week after alcohol exposure began, whereas males were pre-pubertal for most of the two-week exposure period (sex difference in pubertal maturation, *p* < 0.0001; no effect of treatment, *p* > 0.05). (**b**) Alcohol intake (g/kg) within each self-administration bout. Animals had access to sweetened solution (alcohol or water) for 30 minutes for each bout, followed by 60-minute breaks where the lever was retracted (6 bouts/day for two weeks). Both males and females showed evidence of ‘front-loading’ within an overnight session, drinking significantly more during the first bout of each day when compared to the last bout (*p* < 0.01 for both males and females). (**c**) Average g/kg daily alcohol intake (line graph, with shaded SEMs) and total g/kg alcohol intake (bar graph inset) did not differ with sex (*p* > 0.05). (**d**) Three-dimensional model of a rat brain (adapted from [23], and edited using the Sketchup 3D modeling computer program (Trimble Inc., Sunnyvale, CA, USA)) showing the location of the sections taken for analyses in both experiments (2.2 mm anterior to Bregma). The large red box outlining the Cg1 represents the location in which samples were taken for Experiment 1 (myelinated fiber density, layers II-V, (**e**), while the smaller red box represents the location from which images were taken for Experiment 2 (nodal measurements, layer II/III, (**f**). *Abbreviations: Cg1, anterior cingulate cortex, PrL, prelimbic cortex, IL, infralimbic, SEMs, standard errors of the mean.*

**Figure 2 brainsci-09-00167-f002:**
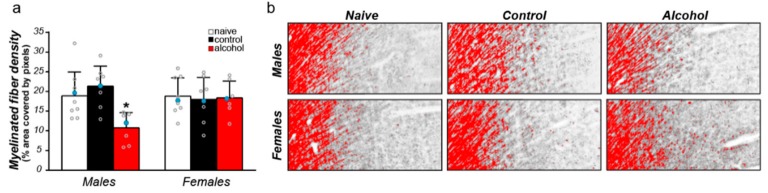
**Adolescent alcohol drinking decreases myelinated fiber density in male, but not female, rats.** (**a**) Adolescent alcohol drinking decreases myelinated fiber density in the Cg1 of males (* *p* = 0.01), with no measurable changes detected in females. Blue circular markers indicate the individuals used for representative images in panel b. *Note:* the fiber density means ± SEM were previously published for males [18]. (**b**) Representative microscopic images from each treatment group, with fibers displayed in red to show density threshold measurements. These representative images were taken with a 5× objective on a light microscope and processed in Image J (open source from https://imagej.net/ImageJ) for enhanced visualization for this figure. Myelin density data were analyzed on scanned images using Aperio ImageScope software (Leica Biosystems Imaging Inc., Vista, CA, USA), as described in the methods.

**Figure 3 brainsci-09-00167-f003:**
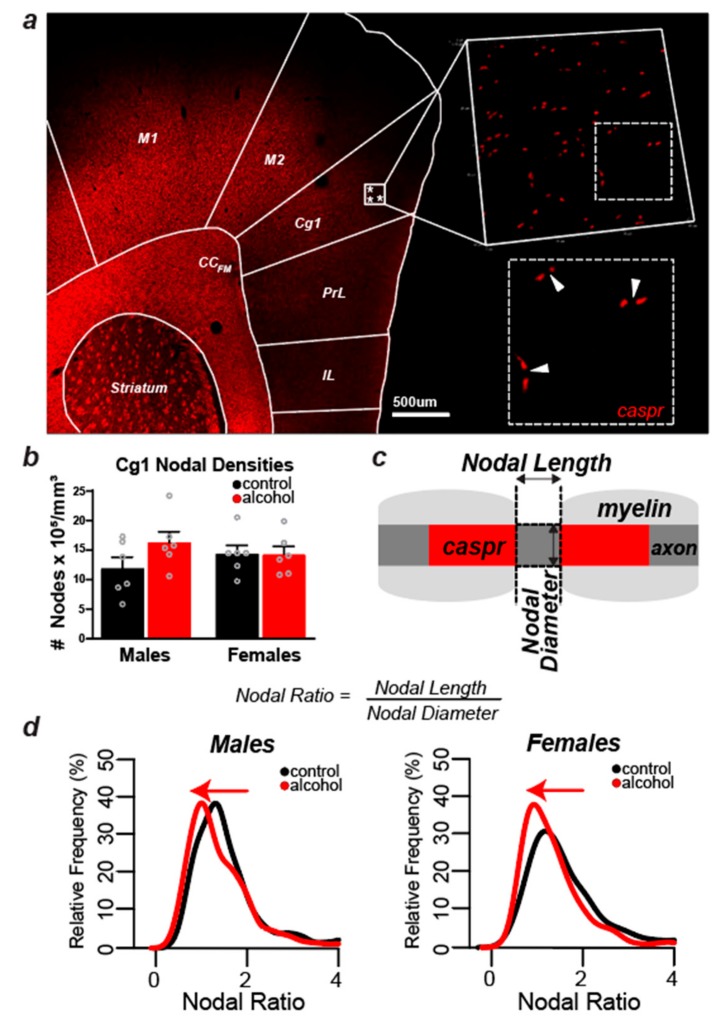
**Nodal Densities and Microstructural Analyses.** (**a**) Coronal slice (10×—stitched image) immunolabeled with contactin-associated protein (Caspr) (in red) to show where samples were taken in layer II/III of the medial prefrontal cortex for nodal analyses. (**b**) No significant changes detected in nodal counts in the medial prefrontal cortex of males and females, although there was a small trend for nodal counts to increase with alcohol consumption in males. (**c**) Schematic drawing to demonstrate how nodal ratio measurements were obtained and calculated. The nodal length was defined as the distance between two corresponding Caspr signals, and the nodal diameter was defined as the width of the positive Caspr signal. Nodal lengths and widths were measured to calculate nodal ratios (nodal length/nodal diameter) of nodes in layers II/III of the Cg1 for all 24 animals (*n* = 6/sex/treatment). For each animal, 40 total nodes were measured (average 10 nodes per z-stack from four optical fields of view). (**d**) Alcohol drinking shifted the nodal ratio distribution towards lower ratios in both males and females (two-sample Kolmogorov–Smirnov test, *p* = 0.001).

**Figure 4 brainsci-09-00167-f004:**
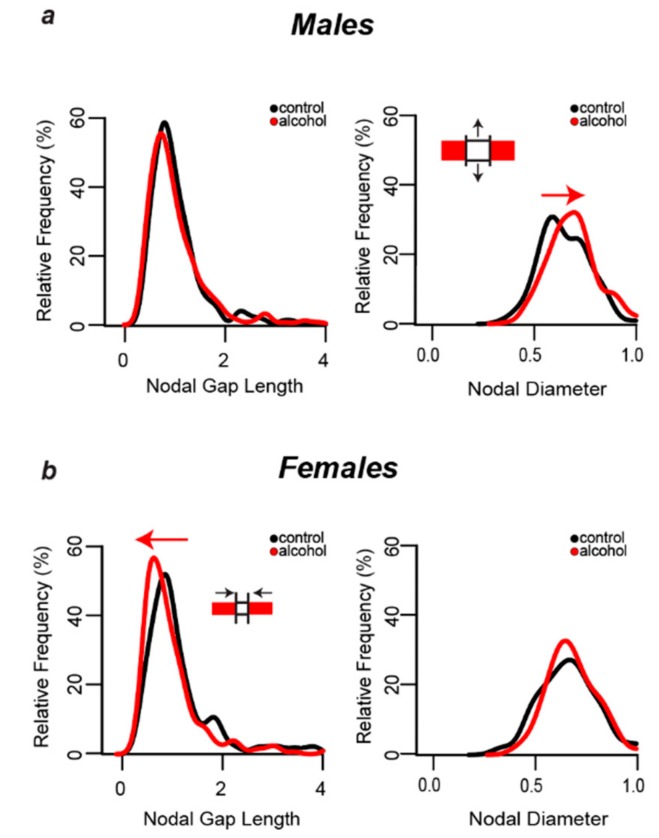
**Sex-Dependent Changes in Nodal Dimensions.** (**a**) Alcohol did not change the nodal gap length distribution in males, but it caused a shift towards larger axonal diameters (two-sample Kolmogorov–Smirnov test, *p* = 0.0004). (**b**) In contrast, alcohol shifted the nodal length distribution in females towards shorter nodes (two-sample Kolmogorov–Smirnov test, *p* = 0.0002), with no change in axon diameter distribution.

**Figure 5 brainsci-09-00167-f005:**
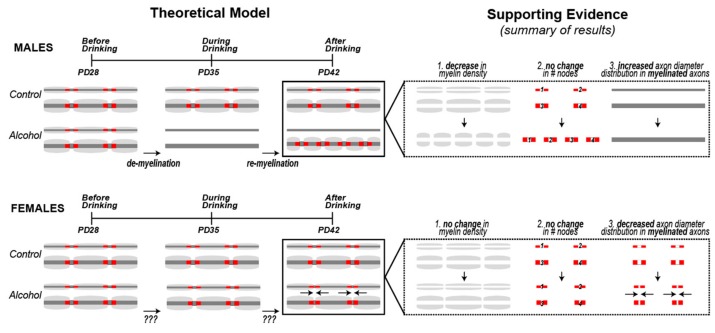
**Theoretical Model Supported by Results Summary.** We propose a model where alcohol causes demyelination, followed by the initiation of remyelination of axons in the Cg1 of males. Three pieces of evidence support this demyelination + remyelination hypothesis in males: (1) a decrease in overall myelin density, coupled with (2) no change in the number of nodes, and (3) a shift in the distribution towards myelinated axons with larger diameters. All three results are consistent with having thinner and shorter myelin sheaths, which is a staple of remyelination [45,46]. We did not observe the same phenomenon in females, suggesting a less robust effect of alcohol on myelin in females, or possibly a more effective remyelination process that returns the myelinated fiber density levels back to normal by the end of the alcohol exposure period.

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
