# Peer review of "Sex Differences in the Effect of Alcohol Drinking on Myelinated Axons in the Anterior Cingulate Cortex of Adolescent Rats"

_brainsci, 2019, doi:10.3390/brainsci9070167_

Round 1

Reviewer 1 Report

Comments to Authors:

This manuscript nicely describes the effects of voluntary alcohol self-administration in adolescent male and female rats on myelin and nodal characteristics in the mPFC. This work is an important addition to the field since it is one of the first to describe these effects in voluntary drinking in adolescent rodents while a majority of the work in the field has used non-contingent alcohol delivery.

Minor issues:

In the methods, in Expt. 1, it is not clear that 3 groups of mice were actually used: naïve, ethanol drinking and control (gluc/sacc) drinking. Please explain the groups in the Methods. Otherwise, the reader does not know naïve animals were also used until Figure 2. It is also not clear if a cohort of the original 47 rats were used for the nodal ratios in Expt. 2. Please clarify.

In Methods 2.4, please include the section thickness.

It is important to note that females did not consume more ethanol than males. This is not typically seen in most rodent studies. Could this be due to the training or added sweetener? Is this finding unique to your lab or paradigm? Please add some discussion about this finding.

The authors provide a mechanism describing the effects of adolescent binge drinking on nodal diameter, length and resulting in lowered nodal ratios. Is anything known about preferential loss of myelin on small axons following alcohol or any type of damage?

On line 250, in the figure legend, the authors state: the data from the males was adapted from [18].” Does this refer to only the image, or the actual data for male Black Gold II staining?

In the references, each reference is double numbered.

Author Response

1. In the methods, in Expt. 1, it is not clear that 3 groups of mice were actually used: naïve, ethanol drinking and control (gluc/sacc) drinking. Please explain the groups in the Methods. Otherwise, the reader does not know naïve animals were also used until Figure 2. It is also not clear if a cohort of the original 47 rats were used for the nodal ratios in Expt. 2. Please clarify. 

Our Response:

The experimental groups have been clarified in the abstract and methods section (2.2). The rationale for the decision to use a subset of the animals for the second experiment and exclude naïve animals has also been provided in this section. “As the naive group was not different from the control group in overall myelin density readout, this group was not included in Exp. 2 as a means of refining the labor-intensive nodal analyses. 

2. In Methods 2.4, please include the section thickness. 

Our Response:

Tissue was sectioned at a 35µm thickness. We have added this information to the methods section (2.4).  

3. It is important to note that females did not consume more ethanol than males. This is not typically seen in most rodent studies. Could this be due to the training or added sweetener? Is this finding unique to your lab or paradigm? Please add some discussion about this finding. 

Our Response:

The reviewer is correct that in most rodent studies females drink more than males, but this sex difference emerges later in adolescence. We have added the following text to results section (3.2): The lack of sex differences at this early adolescent time period is a consistent finding in our lab, which is likely because sex differences emerge later in adolescence [28] and continue into adulthood [29,30]. 

4. The authors provide a mechanism describing the effects of adolescent binge drinking on nodal diameter, length and resulting in lowered nodal ratios. Is anything known about preferential loss of myelin on small axons following alcohol or any type of damage? 

Our Response: 

This is a great question. We have provided additional information and edited the discussion to address this: “We also found that after alcohol, the population of myelinated axons shifted toward larger diameter axons. These results could indicate preferential demyelination of smaller axons, or alternatively preferential remyelination of larger axons following demyelination. While it is currently unknown whether smaller axons are more vulnerable to demyelination, it is known that larger axons are remyelinated first [42,45]. Regardless of the details of the demyelinating/remyelinating processes, the consequential changes in myelin microstructure could have major functional consequences, as shorter internodes can result in decreased conduction velocity [46].” 

5. On line 250, in the figure legend, the authors state: the data from the males was adapted from [18].” Does this refer to only the image, or the actual data for male Black Gold II staining? 

Our Response:  

The Black Gold II myelinated fiber density data for males was published in Vargas et al, 2014. The journal gives permission to re-publish the data as long as it is referenced. The bar graph is similar to the previous publication but we provided additional information by including dots for individualized data points and images of representative examples of fiber density. To clarify this, we have added the following sentence to the figure legend: Note: the fiber density means ± S.E.M. were previously published for males [18]. 

6. In the references, each reference is double numbered. 

Our Response:This error has been corrected. 

Reviewer 2 Report

In the manuscript “Sex differences in the effect of alcohol drinking on myelinated axons in the anterior cingulate cortex of adolescent rats”, Tavares et al. analyzed myelinated fiber density and microstructural features of the Nodes of Ranvier in the anterior cingulate cortex (layers II/III) of rats following voluntary alcohol consumption during early adolescence. They reported a sex-dependent effect of alcohol on myelinated fiber density and nodal microstructures. The findings are fairly straightforward and contribute to the understanding of mechanisms underlying cognitive associated with teenage drinking.  A few requests for clarification are noted below.

1.     Rationale for focusing the layer II/III of the anterior cingulate cortex in nodal analysis should be included considering that analysis of myelin density was performed in layers II-V. Is it possible that the observed myelin density reduction was driven by layer V while the density remained unchanged in layer II/III?

2.     There is some confusion about the number of z-stack images per animal in the analysis. The authors may need to clarify the use four images per animal for nodal count measurements but all images (4-6) of each animal for nodal ratio measurements. In line 185, the authors then mentioned “Approximately 10 nodes per Cg1 z-stack region were analyzed for a total of 40 nodes per animal” while it seems not possible considering some animals may have 6 images and therefore more nodes than animals with only 4 images. This detail matters because it may affect the nodal ratio distribution that the distribution can be driven by animals who had more nodes assessed.

3.     The author reported shifted nodal ratio distribution and sex-specific alterations in nodal domain measurements. Is there a sex difference in control animals? The distribution looks similar between alcohol-exposed males and females. It would be interesting to see if alcohol exposure during adolescence reduced the sex dimorphism in Cg1.

4.     In the discussion, the authors stated that “the observed shift to smaller nodal gap lengths in females could be indicative of a compensatory mechanism to combat successive binge drinking episodes, or rather, this shift could imply impaired functionality”. The discussion could be enhanced by including a brief discussion about whether previous studies observed cognitive deficits associated with Cg1 in females following adolescence alcohol exposure, and whether the cognitive deficits differ to males.

Author Response

1.     Rationale for focusing the layer II/III of the anterior cingulate cortex in nodal analysis should be included considering that analysis of myelin density was performed in layers II-V. Is it possible that the observed myelin density reduction was driven by layer V while the density remained unchanged in layer II/III? 

OUR RESPONSE:  

    The rationale for focusing on layer II/III of the anterior cingulate cortex in the nodal analysis has been added to the methods section (2.6.2): “We limited the sampling area to these layers to overcome the technical and labor-intensive challenges of this analysis. We have initially focused on layer II/III because the basolateral amygdala is still sending afferents to these layers during adolescence and into early adulthood [25], potentially making these developing fibers more vulnerable to the effects of alcohol.” 

     In addition, to address the point that myelin density reduction may be driven by layer V while remaining unchanged in layer II/III, the following text has been added to the discussion: “One caveat of our study is that the myelin density changes observed in males includes layers II-V, but only layer II/III was analyzed at the microstructural level. While our findings indicate that alcohol alters nodal domains of myelinated axons within layer II/III of males and females, it is also possible that alcohol causes similar microstructural changes to axons in layer V.  This is especially important because this drinking period overlaps with de novo myelination of the axons extending from the CCFM through layer V of the Cg1, which is accompanied by a doubling of transmission speed [41].” 

2.     There is some confusion about the number of z-stack images per animal in the analysis. The authors may need to clarify the use four images per animal for nodal count measurements but all images (4-6) of each animal for nodal ratio measurements. In line 185, the authors then mentioned “Approximately 10 nodes per Cg1 z-stack region were analyzed for a total of 40 nodes per animal” while it seems not possible considering some animals may have 6 images and therefore more nodes than animals with only 4 images. This detail matters because it may affect the nodal ratio distribution that the distribution can be driven by animals who had more nodes assessed. 

OUR RESPONSE:  

    For the analysis in which the number of nodes is counted, exactly 4 z-stacks (all containing the same volume) were used for each animal. However, for further analyses of the nodal dimensions, we sampled 40 nodes per animal. In some cases, a 5th or even 6th z-stack was needed in order to obtain 40 nodal measurements in total. To clarify this, we have these details and the following text to the methods section: “a total of 40 nodes were analyzed per animal, obtained from 4-6 z-stacks per animal (the average number of z-stacks did not differ across groups).” 

  

3.     The author reported shifted nodal ratio distribution and sex-specific alterations in nodal domain measurements. Is there a sex difference in control animals? The distribution looks similar between alcohol-exposed males and females. It would be interesting to see if alcohol exposure during adolescence reduced the sex dimorphism in Cg1. 

OUR RESPONSE: 

    There were no sex dimorphisms in nodal ratios (p=0.15), length (p=0.38) or diameter (p=0.22) measurements in control animals. We have added this information to the results section. 

4.     In the discussion, the authors stated that “the observed shift to smaller nodal gap lengths in females could be indicative of a compensatory mechanism to combat successive binge drinking episodes, or rather, this shift could imply impaired functionality”. The discussion could be enhanced by including a brief discussion about whether previous studies observed cognitive deficits associated with Cg1 in females following adolescence alcohol exposure, and whether the cognitive deficits differ to males. 

OUR RESPONSE: 

    This is a really important point to discuss; accordingly, we have added the following text to the discussion section (end of the 4th paragraph): “Indeed, exposure to high doses of alcohol via gavage can elicit working memory deficits in adult male and female rats [52]. We have found that high adolescent drinking levels predicted working memory deficits in adulthood in males [18]. It remains to be determined if this relationship is also observed in high drinking females.”